# The Length of Residence is Associated with Cardiovascular Disease Risk Factors among Foreign-English Teachers in Korea

**DOI:** 10.3390/bs8010002

**Published:** 2017-12-26

**Authors:** Brice Wilfried Obiang-Obounou

**Affiliations:** Department of Food Nutrition, College of Natural Sciences, Keimyung University, 1095 Dalgubeoldaero, Dalseo-gu, Daegu 42601, Korea; obiang@gw.kmu.ac.kr; Tel.: +82-53-580-5780

**Keywords:** cardiovascular disease, diabetes, hypertension, acculturation

## Abstract

Cardiovascular disease (CVD) is a group of disorders that involve the heart and blood vessels. Acculturation is associated with CVD risk factors among immigrants in Western countries. In this study, the association between acculturation and CVD risk factors was examined among English teachers from Europe and the USA living in Korea. English teachers were defined as those who reported their profession as “English Teacher”. Only English teachers from Europe (UK, and Ireland, *n* = 81) and North America (Canada and USA, *n* = 304) were selected. The length of residence and eating Korean ethnic food were used as proxy indicators for acculturation. Gender was associated with hypertension: 17.6% of males self-reported to have the cardiovascular risk factor when compared to females (7.4%). The length of residence in Korea was associated with hypertension (*p* = 0.045), BMI (*p* = 0.028), and physical inactivity (*p* = 0.046). English teachers who had been residing in Korea for more than five years were more likely to report hypertension (OR = 2.16; *p* = 0.011), smoking (OR = 1.51; *p* = 0.080), and overweight/obesity (OR = 1.49; *p* = 0.009) than participants who had been living in Korea for less than five years. This study found evidence of the healthy immigrant effect and less favorable cardiovascular risk profiles among English teachers who have lived in Korea for over five years.

## 1. Introduction

Cardiovascular disease (CVD) is a group of disorders that involve the heart and blood vessels. With an estimated 17.7 million people dying from CVDs in 2015—representing 31% of all global deaths—CVD is now the leading cause of mortality worldwide [1]. While “non modifiable” risk factors such as age, gender, and family history cannot be changed, other risk factors such as hypertension (high blood pressure), diabetes, smoking, being overweight/obese, being inactive, and bad diet can be changed. The World Health Organization (WHO) reported that high blood pressure resulted in 13% of CVD deaths, while tobacco resulted in 9%, diabetes 6%, lack of exercise 6%, and obesity 5% [2]. Other than these risk factors, scholars have also identified immigration as a predictor of cardiovascular risk through the theory known as the “healthy immigrant effect” where it has been proven that newly arrived immigrants have lower rates of cardiovascular risk factors than nonimmigrants [3,4]. As new immigrants arrive in the host country, they go through a cultural transition called acculturation.

Acculturation is the process of adapting to the traditions, values and practices of a host country [5]. The length of residence, the most commonly used proxy measure of acculturation, is associated with increased prevalence of CVD risk factors among immigrants in Western countries [1,3,6,7]. Little is known about the association between the length of residence of immigrants and CVD risks among new immigrants in the Republic of Korea (Korea) as most research has mainly focused on the acculturation of Hispanic, European, and Asian immigrants in Western societies [8,9]. Korea was mostly known for its large-scale emigration, however it has recently emerged as a popular destination for immigrants [10,11]. For the first time, it is estimated that three out of every 100 individuals in Korea originated from a foreign nation [12,13]. The country is consequently transitioning from a homogenous society to a multicultural country” [11,13,14]. Until very recently, Korea has had little or no experience with large-volume immigration [8]. As a tool for advancement or survival [15], the Korean government welcomes a great number of English teachers mainly from seven countries: Canada, USA, UK, Australia, New Zealand, and South Africa.

Given the rapid increase in English teachers mainly from Canada and the US (around 5000 in public schools), this study aimed to examine the association between the duration of residence and CVD risk factors mediated by gender and the dietary acculturation of English teachers in Korea. To my knowledge, this is first research studying acculturation and CVD risk factors in Korea. The preliminary conceptualized model of the relationship between the duration of residence in Korea, dietary acculturation, and CVD risk factors is shown in Figure 1.

## 2. Materials and Methods

### 2.1. Study Population and Sample 

A cross-sectional survey was carried out between August and December 2014 among the foreign-born population in Korea. Participants were selected through population-based approaches. In order to have the maximum number of foreigners for the study, posters were placed in public coffee shops, universities, and foreigner based social groups (i.e., Facebook). Before the participants answered the questionnaire, they received information from the researcher about the survey including its purpose, procedures, and confidentiality. The criteria for inclusion were: foreign-born, aged 18 and over, and residing in Korea. The exclusion criteria included aged under 18, participants residing outside of Korea, Koreans, and pregnant women. A total of 914 participants from Africa (236), Asia (140), Europe (125), North America (325), the Middle East (23), Australia (27), and South America (30) completed baseline questionnaires (Figure 2). English teachers were defined as those who reported their profession as “English Teacher”. Only participants from Canada, USA, UK, and Ireland were selected as Australia, New Zealand, and South Africa did not have a statistical representation of the sample (less than 50 people reported their profession as English teachers). The countries selected were then grouped in two categories: Europe (UK and Ireland) and North America (Canada and USA).

#### 2.1.1. Acculturation Variables

Demographic variables included age, sex, education level (high school, bachelor’s degree or post-graduate), marital status, and occupation. The number of years residing in Korea was used as an acculturation proxy. Prior studies have found length of residence in a country to be a valid and reliable proxy of acculturation in Western countries [1,3,6,7]. The duration of residence in Korea was assessed by the question: “How long have you been living in Korea?” [16]. Responses were categorized as a three-level variable (<1 year, 1–4 years, and ≥5 years) [17]. While acculturation pertains to adopting cultural traits, dietary acculturation specifically refers to the process that occurs when members of a minority group adopt the eating patterns/food choices of their new environment [5,18]. Dietary acculturation was assessed by the question: “Do you regularly eat typical Korean food?” Eating typical Korean food was defined by answering “yes or no” to the question.

#### 2.1.2. Cardiovascular Risk Factors 

The study examined five CVD risk factors: hypertension, diabetes, current smoking, overweight/obesity, and physical inactivity. Self-reported hypertension and diabetes were ascertained as previously published [19] by the question: “Has a doctor ever told you that you have high blood pressure/diabetes?” Current smoking was defined by answering “yes” to “Do you currently smoke cigarettes?” Body mass index (BMI) was calculated based on the participant’s self-reported weight and height as weight in kilograms divided by height in meters squared (kg/m^2^). The World Health Organization (WHO) has recommended classifications of bodyweight that include degrees of underweight and gradations of excess weight or overweight that are associated with increased risk of some non-communicable chronic diseases [20,21]. The normal range for an adult BMI is 18.5–24.9; an adult with a BMI ranging from 25–29.9 is considered overweight whereas an adult with a BMI greater than or equal to 30 is considered as obese [22]. Physical inactivity was defined as not meeting the WHO physical activity guidelines [23]. The answer “yes” to the statement “I did less than 150 min of moderate-to-vigorous intensity physical activity (at least 10 min) in the previous week”.

### 2.2. Human Subjects Research Approval

The protocol was approved by the Institutional Review Board (IRB) of Keimyung University (IRB file number: 40525-201410-HR-71-02).

### 2.3. Statistical Analysis

Statistical analyses were conducted using IBM SPSS Statistics for Windows, Version 21.0 (Armonk, NY, USA). Univariate analyses were used to generate descriptive statistics and the Chi-square test was employed. The study used bivariate and multivariate analyses to determine the association between length of residence and CVD risk factors. All p values reported were for 2-tailed tests and a value of *p* < 0.05 was considered statistically significant. Reported odds ratios were based on the final logistic regression model.

## 3. Results

### 3.1. Univariate Analyses

There were 914 self-identified foreign-born participants (Figure 2).

After stratification based on country of origin and profession, 81 participants from Europe (UK and Ireland) and 304 from North America (Canada and USA) were selected for the study (Table 1). Other countries (continents) were excluded from two reasons. First, only English native speakers are allowed to teach English in Korea. Second, countries (continents) with less than 10 participants as English teachers were not significant. The average age of the sample was 33.23 years (SD = 8.50), 56% were women, and over half had lived in Korea for less than five years. The sample was composed of highly educated participants with only 2.1% said not to have completed a bachelor degree.

### 3.2. Cardiovascular Risk Factors

#### 3.2.1. Gender and CVD Risk Factors 

Gender was associated with hypertension: 17.6% of males self-reported to have the cardiovascular risk factor when compared to females (7.4%) (Table 2). Furthermore, a positive correlation was also observed between gender and BMI (*p* = 0.003) with 47.4% of the participants being overweight/obese. With 63.3% of the participants not meeting the WHO physical activity guidelines for physical activity, the strong association observed between gender and physical inactivity (*p* = 0.001) could be mostly attributed to the 70.3% of the female participants reporting not meeting the WHO guidelines. No association was observed between gender and diabetes and smoking. However, two of the three cases of diabetes and over 65% of smokers were male participants. Typical Korean food was not a factor between gender as over 95% of both females and males had typical Korean food as part of their diet.

#### 3.2.2. Length of Residence and CVD Risk Factors

The length of residence in Korea was associated with hypertension (*p* = 0.045), BMI (*p* = 0.028), and physical inactivity (*p* = 0.046) (Table 3). There was no association with diabetes. Diabetes only featured in three cases and was not enough to determine a correlation. The absence of association between a longer residence in Korea is that smoking is habit often acquired during the teenage years. English teachers in their majority incorporated Korean diet into their food habits from their first contact with the host country.

#### 3.2.3. Length of Residence in Multivariate Analyses

The final multivariate logistic regression model (Table 4) included hypertension, diabetes, smoking, overweight/obesity, and physical inactivity. Since over half of the participants had lived in Korea for less than five years, the length of residence was then grouped into two categories: “<5 years” and “>5 years”. As expected, the final model showed no association with diabetes and typical Korean food. On the other hand, those who had been residing in Korea for more than five years were more likely to report hypertension (OR = 2.16; *p* = 0.011), smoking (OR = 1.51; *p* = 0.080) and overweight/obesity (OR = 1.49; *p* = 0.009) than participants who had been living in Korea for less than five years. However, participants reporting not meeting the WHO guidelines for physical activity were 46% less likely to meet them the longer they resided in Korea (OR = 0.54; *p ≤* 0.0001).

## 4. Discussion

Many studies have found a correlation between acculturation and CVD risk factors in Western countries [1,3,6,7]. This was the first study to examine the association between duration of residence and CVD risks factors mediated by gender and dietary acculturation of English teachers in Korea. Given the small sample used (*n* = 385), the fact that more than 11% reported as currently smoking and having hypertension should raise alarms about the health status of this growing population in Korea. Furthermore, overweight/obesity (47.4%) and the lack of physical activity (63.3%) are two known risk factors of cardiovascular disease [24,25].

Gender is a “non-modifiable” risk factor and cannot be changed. Although CVD remains the leading killer of both women and men worldwide, there are substantial gender differences in the prevalence of different CVDs [26]. Even though more women than men die from CVD every year, this group of diseases has often been viewed as a man’s disease [26,27]. When it comes to CVD risk factors, the results are inconsistent as race is often a factor. In the U.S., for example, the highest rate of hypertension is among black women when compared to other races [26,28]. Our study did not differentiate race; however, the findings of this study were consistent with other studies showing that men had higher levels of hypertension [29,30]. Observed gender differences in hypertension could be behavioral [30,31]; with BMI [30,32], smoking [30,33], and physical inactivity [30,32] as risk factors. The examination of Table 2 shows that CVD risk factors were predominant in men with 16.5% of men who were current smokers when compared to 6.9% for women. In addition, 47.4% of men were overweight/obese when compared to women (41.7%).

This study was in agreement with several lines of evidence that have shown that the prevalence of obesity and hypertension increased with duration of residence in the host country [7,34]. Obesity is a serious public health epidemic that this century has known; there are approximately 2 billion adults that are overweight or obese. Prior studies have found that immigrants from low- to medium-income countries who have migrated to and reside in high-income countries are more susceptible to overweight and obesity than their local counterparts [35,36]. This study showed that the phenomenon was the same from high-income countries (USA, Canada, UK, and Ireland) to another high-income country (Korea). Furthermore, among the US and Australian based migrant groups, people born in Italy, Greece or Cyprus, the former Yugoslavia, Germany, the Netherlands, Poland and other Eastern European countries recorded a higher likelihood of overweight and obesity than people born in Malaysia, Vietnam or Cambodia, the Philippines, or China [37].

Although studies have suggested that longer residence is associated with diabetes and smoking, an increasing length of residence in Korea was not significantly associated with diabetes and smoking. Only three English teachers were reported to have diabetes. This number is not sufficient to determine a correlation. Furthermore, even though two of the three reported cases of diabetes were from participants living in Korea for over five years, the lower reporting of diabetes as a CVD risk factor could be associated to the “healthy immigrant effect” [3,4].

A decrease in physical inactivity with increasing length of residence in Korea was also observed. There are two possible reasons for this inverse association. First, the English teachers may have been aware of their health condition and consequently decided to exercise more. Second, as stated by Koya, this may have been part of a “positive acculturation effect” [6]. As immigrants reside longer in the host country and acculturate to their new environment, they are more likely to adopt healthy beliefs on the benefits of exercise.

After stratification for the length of residence in Korea to <than 5 years and >5 years, the odds of smoking became 1.5 greater for participants residing in Korea for over five years (Table 4). However, it is possible that the habit could have been acquired during teenage years before migrating to Korea and not necessary attributed to acculturation. The average age of English teachers in this study was 33.23 years (SD 8.50). Out of the 43 people who reported to be smokers, twenty had lived in Korea for less than five years when compared to the 23 said to be in Korea for over five years. Nevertheless, smoking is a modifiable CVD risk factor and awareness programs aimed at English teachers in Korea are necessary.

Finally, typical Korean food was not associated with CVD risk factors as over 95% of English teachers in Korea easily went through a nutrition transition. Nutritional transition relates to the tendency in decreased consumption of healthy and nutritional foods in favor of fatty and processed foods [38,39]. The weight gain observed among English teachers was likely to be associated to poor diet choice. Korean food is generally regarded as healthy although the country has been going through a diet transition [40,41,42]. Concurrent changes in lifestyle include the rapid introduction of elements of what may be termed a Western lifestyle despite national efforts to retain elements of a traditional diet. It is therefore possible that English teachers in Korea will choose Korean food with more familiar elements as food habits are likely to be the last to change when people become acculturated.

### Limitations

This study had some limitations. The majority of participants were well educated (Table 1), which limits generalization. Data on hypertension and diabetes were self-reported, which may be subject to recall bias or prone to misclassification. However, self-reported data of hypertension have been shown to be highly correlated with medical records. The results may have also been influenced by a potential cohort effect since the study did not distinguish between pre- and post-migration diagnosis of hypertension or diabetes by a medical doctor. Since self-reported hypertension was associated with duration of residence in Korea, measuring pre- and post-migration hypertension by a medical doctor would provide a more accurate association. On the other hand, the fact that the findings of this study were consistent with previous studies [7,29,30,34] lends evidence to the face validity of this study. The length of residence and eating Korean ethnic food were used as proxy indicators for acculturation. Other measures of acculturation such as language proficiency, participating in ethno-cultural networks, and celebrating holidays and cultural events were not examined. Future studies with larger samples and race specifications are needed since there might be heterogeneity within English teachers in Korea. This will provide a better understanding of how acculturation influences cardiovascular risk factors among immigrants in Korea.

## 5. Conclusions

Korea has been composed of a single ethnic group with no experience of large scale immigration. This study found evidence of the healthy immigrant effect and less favorable cardiovascular risk profiles among English teachers who had lived in Korea for over five years. Given that the Korean government welcomes a great number of English teachers every year, there is a need to implement awareness programs targeting CVD risk factors and promoting a healthier lifestyle for these newcomers. The same programs could be applied to other groups of immigrants such as foreign women married to Korean men as this group also face health challenges [43].

## Figures and Tables

**Figure 1 behavsci-08-00002-f001:**
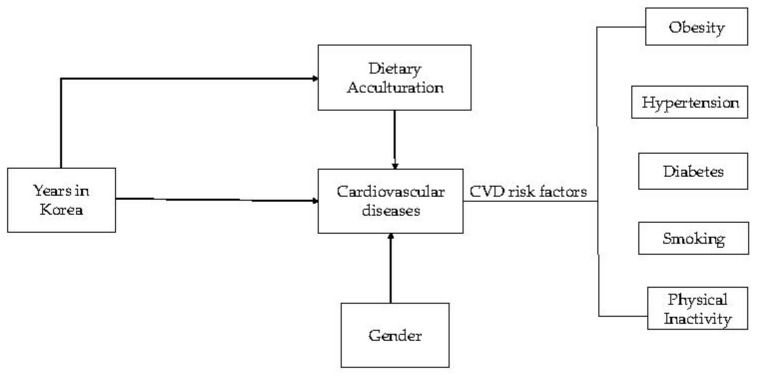
Preliminary conceptual model of the relationship between duration of residence in Korea, dietary acculturation, and cardiovascular disease (CVD) risk factors.

**Figure 2 behavsci-08-00002-f002:**
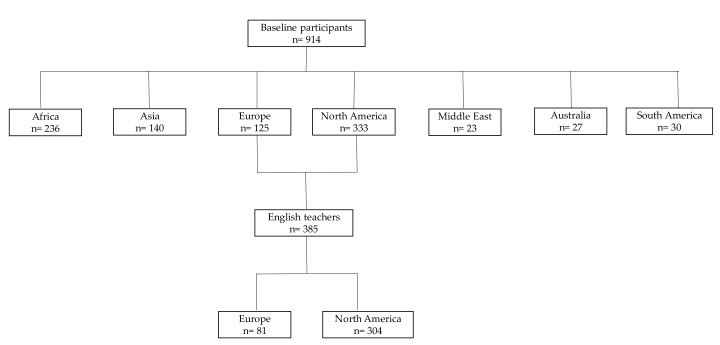
Participants flow diagram.

**Table 1 behavsci-08-00002-t001:** Sociodemographic characteristics.

	Total (*n* = 385)	Europe (*n* = 81)	North America (*n* = 304)
Age at interview (years)			
Mean (SD)	33.23 (8.50%) ^1^	33.33 (8.36%)	33.20 (8.55%)
Range	20–68	22–63	20–68
Gender			
Female	216 (56.0%)	31 (38.3%)	185 (60.7%)
Male	170 (44.0%)	50 (61.7%)	120 (39.3%)
Education			
High School	8 (2.1%)	3 (3.7%)	5 (1.6%)
Undergraduate	196 (50.8%)	36 (44.4%)	160 (52.5%)
Post-graduate	182 (47.2%)	42 (51.9%)	140 (45.9%)
Marital status			
Single	219 (56.7%)	40 (49.4%)	179 (58.7%)
Married	167 (43.3%)	41 (50.6%)	126 (41.3%)

^1^ Column percentage are shown in parentheses.

**Table 2 behavsci-08-00002-t002:** Association between gender and cardiovascular risk factors.

CVD Risk Factors	Total	Female	Male	*p* Value
Hypertension				
Yes	46 (11.9%) ^1^	16 (7.4%)	30 (17.6%)	0.002
Diabetes				
Yes	3 (0.8%)	1 (0.5%)	2 (1.2%)	0.433
Smoking				
Yes	43 (11.1%)	15 (6.9%)	28 (16.5%)	0.003
Body Mass Index (kg/m^2^)				
Underweight	5 (1.3%)	4 (1.9%)	1 (0.6%)	
Normal	198 (51.3%)	122(56.5%)	76 (44.7%)	0.003
Overweight/obese	183 (47.4%)	90 (41.7%)	93 (47.4%)	
Physical Inactive				
Yes	235 (63.3%)	147 (70.3%)	88 (54.3%)	0.001
Typical Korean diet				
Yes	328 (96.5%)	187 (95.9%)	141 (97.2%)	0.574

^1^ Column percentage are shown in parentheses.

**Table 3 behavsci-08-00002-t003:** Association between length of residence and cardiovascular risk factors.

	Duration of Residence in Korea				
CVD risk factors	Total	<1 year	1 to 4 years	Over 5 years	*p* value
Hypertension					
Yes	46 (11.9%)	6 (9.8%)	13 (8.0%)	27 (16.7%)	0.046
Diabetes					
Yes	3 (0.8%)	0 (0.0%)	1 (0.6%)	2 (1.2%)	0.621
Smoking					
Yes	43 (11.1%)	5 (8.2%)	15 (9.2%)	23 (14.2%)	0.262
Body Mass Index (kg/m^2^)					
Underweight	5 (1.3%)	3 (4.9%)	1 (0.6%)	1 (0.6%)	0.028
Normal	198 (51.3%)	34 (55.7%)	89 (54.6%)	75 (46.3%)	
Overweight/obese	183 (47.4%)	24 (39.3%)	73 (44.8%)	86 (53.1%)	
Physical Inactive					
Yes	235 (63.3%)	39 (63.9%)	112 (72.3%)	84 (54.2%)	0.046
Typical Korean diet					
Yes	328 (96.5%)	48 (94.1%)	137 (96.5%)	143 (97.3%)	0.571

**Table 4 behavsci-08-00002-t004:** Multivariate logistic regression: association between length of residence and CVD risk factors.

CVD Risk-Factors	Length of Residence	Adjusted OR (95%CI)	*p* Value
Hypertension	<5 years	Ref	
	Over 5 years	2.16 (1.18–3.96)	0.011 *
Diabetes	<5 years	Ref	
	Over 5 years	4.33 (0.45–41.80)	0.168
Smoking	<5 years	Ref	
	Over 5 years	1.51 (0.95–2.41)	0.080 **
Overweight/Obesity	<5 years	Ref	
	Over 5 years	1.49 (1.11–2.01)	0.009 ***
Physical Inactivity	<5 years	Ref	
	Over 5 years	0.54 (0.40–0.74)	0.000 ***
Korean typical diet	<5 years	Ref	
	Over 5 years	1.47 (0.59–3.69)	0.410

* *p* < 0.05; ** *p* < 0.01, *** *p* < 0.001.

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
