# Peer review of "The Length of Residence is Associated with Cardiovascular Disease Risk Factors among Foreign-English Teachers in Korea"

_behavsci, 2017, doi:10.3390/bs8010002_

Round 1

Reviewer 1 Report

This manuscript reports on factors associated with cardiovascular diseases among immigrant English Teachers residing in Korea. Two proxy of acculturation is used, namely time of residence in host country, and consumption of Korean food. The introduction is lacking  evidence of   what is known on this  topic in Korea. Furthermore, the findings of the study for the most part  the immigrant health effect. However, there are some contradictions in the results and discussion sections. These issues make it unclear as to the overall direction of the paper. 

Author Response

Dear Reviewer,

Thank you for taking the time to review my manuscript. The followings are my answers to your questions

Answers to reviewer 1

This manuscript reports on factors associated with cardiovascular diseases among immigrant English Teachers residing in Korea. Two proxy of acculturation is used, namely time of residence in host country, and consumption of Korean food.

The introduction is lacking evidence of   what is known on this topic in Korea.

This is a great observation. Korea was mostly known for its large-scale emigration, however it has recently emerged as a popular destination for immigrants 1-2. For the first time, it is estimated that three out of every 100 individuals in Korea originated from a foreign nation 3-4. The country is consequently transitioning from a homogenous society to a multicultural country 2, 4-5.

This upper section was added to the introduction.

Furthermore, the findings of the study for the most part the immigrant health effect. However, there are some contradictions in the results and discussion sections. These issues make it unclear as to the overall direction of the paper. 

I agree with the reviewer in this assessment and the correction was made. The migrant health effect or healthy migrant effect means that the newly arrived migrants tend to be healthier than the host population. This study found evidence of the healthy immigrant effect and higher cardiovascular risk profiles among English teachers who have lived in Korea for over five years.

1.          Kim, S., Soft talk, hard realities: Multiculturalism as the South Korean government's decoupled response to international migration. Asian and Pacific Migration Journal 2015, 24 (1), 51-78.

2.          Im, H.; Lee, K. Y.; Lee, H. Y., Acculturation stress and mental health among the marriage migrant women in Busan, South Korea. Community Ment Health J 2014, 50 (4), 497-503.

3.          Doo-Sub, L. S. K., Acculturation and Self-rated Health among Foreign Women in Korea. Health and Social Welfare Review 2014, 34 (2 ), 453-483.

4.          Jun, H.-J.; Ha, S.-K., Social capital and assimilation of migrant workers and foreign wives in South Korea: The case of Wongok community. Habitat International 2015, 47, 126-135.

5.          Kim, M.; Park, G. S.; Windsor, C., Marital power process of Korean men married to foreign women: A qualitative study. Nursing & Health Sciences 2013, 15 (1), 73-78.

Reviewer 2 Report

In this paper the effect of acculturation on CVD risk factors in Korea is analyzed. Although the paper has some limitations, as the author highlight, the paper has the value of being carried with immigrants in Korea. I think the topic of this manuscript is of interest for the readers of Behavioral Sciences. However, in my opinion the author could improve the manuscript by:

 Page 2, line 48: How many English teachers are in Korea?

Study population: Why English teachers are selected? The inclusion criteria do not include English teacher profession.

 P2,L66: The sum by countries is 906, not 914. North Amercian = 333

P2,L69: Please, better explain why Australia, New Zealand and South Africa are excluded from the analysis since country of origin is not included in the conceptual model.

 Acculturation variables: age, education level, marital status and occupation are not included in the conceptual model. Also, if all participants are English teachers, occupation should be the same for all the included participant.

P3,L82: The possible responses are Yes or No? 

P3: Delete lines 109-111

P3, L114-115: Delete sentence since the number of participants by country of origin was given in L66

P4,L118: It should say Figure 2.

Figure 2: Include reasons for exclusion.

Table 1: include years in Korea and eat Korean food.

Tables 2 and 3: include number of missing values if any.

Discussion: In my opinion, the author could highlight that this is one of the few or the first paper studying acculturation and CVD risk factors in Asian countries.

Discussion: How is the trend of immigration in Korea?

Author Response

Dear reviewer,

thank you very much for your valuable comments. the manuscript is better because of your suggestions. Below are my responses.

Answers to reviewer 2

In this paper the effect of acculturation on CVD risk factors in Korea is analyzed. Although the paper has some limitations, as the author highlight, the paper has the value of being carried with immigrants in Korea. I think the topic of this manuscript is of interest for the readers of Behavioral Sciences. However, in my opinion the author could improve the manuscript by:

 Page 2, line 48: How many English teachers are in Korea?

There is no official number. As of 2016, it is estimated that the number of English teachers in Korea is around 5,000 in public schools only.

Study population: Why English teachers are selected? The inclusion criteria do not include English teacher profession.

 P2,L66: The sum by countries is 906, not 914. North Amercian = 333

The total number is 914 ( Africa 236, Asia 140, Europe 125, 333, Middle East 23, Autralia 27, South America 30)

P2,L69: Please, better explain why Australia, New Zealand and South Africa are excluded from the analysis since country of origin is not included in the conceptual model.

The country of origin was included. However, it was later categorized into continents. From these continents, only North America and Europe a sample for acceptable statistical analysis. Consequently Asutralia and New Zeland were excluded as only participants were English teachers.

 Acculturation variables: age, education level, marital status and occupation are not included in the conceptual model. Also, if all participants are English teachers, occupation should be the same for all the included participant.

A total of 385 participants were retained for the study as they reported “english teacher” as their profession in Korea

P3,L82: The possible responses are Yes or No? 

Thank you for the observation. Yes the responses were yes or no. The following sentence was added to that specific question: “Eating typical Korean food was defined by answering “yes or no” to the question.” P3L89

P3: Delete lines 109-111

Thank you, lines deleted

P3, L114-115: Delete sentence since the number of participants by country of origin was given in L66

Thank you, Lines deleted

P4,L118: It should say Figure 2.

Change made

Figure 2: Include reasons for exclusion.

The following lines were added to the text: “ Other countries (continents) were excluded from two reasons. First, only English native speakers are allowed to teach English in Korea. Second, countries (continents) with less than 10 participants as English teachers were not significant”.

Table 1: include years in Korea and eat Korean food.

These can’t be added since table 1 is summary of sociodemographic sample.

Tables 2 and 3: include number of missing values if any.

No missing values

Discussion: In my opinion, the author could highlight that this is one of the few or the first paper studying acculturation and CVD risk factors in Asian countries.

Thank you very much. The following truthful statement was added: “To my knowledge, this is first research studying acculturation and CVD risk factors in Korea”.

Discussion: How is the trend of immigration in Korea?

I added the followings to the introduction as the same question was asked by another reviewer:

“ Korea was mostly known for its large-scale emigration, however it has recently emerged as a popular destination for immigrants 1-2. For the first time, it is estimated that three out of every 100 individuals in Korea originated from a foreign nation 3-4. The country is consequently transitioning from a homogenous society to a multicultural country” 2, 4-5.

1.          Kim, S., Soft talk, hard realities: Multiculturalism as the South Korean government's decoupled response to international migration. Asian and Pacific Migration Journal 2015, 24 (1), 51-78.

2.          Im, H.; Lee, K. Y.; Lee, H. Y., Acculturation stress and mental health among the marriage migrant women in Busan, South Korea. Community Ment Health J 2014, 50 (4), 497-503.

3.          Doo-Sub, L. S. K., Acculturation and Self-rated Health among Foreign Women in Korea. Health and Social Welfare Review 2014, 34 (2 ), 453-483.

4.          Jun, H.-J.; Ha, S.-K., Social capital and assimilation of migrant workers and foreign wives in South Korea: The case of Wongok community. Habitat International 2015, 47, 126-135.

5.          Kim, M.; Park, G. S.; Windsor, C., Marital power process of Korean men married to foreign women: A qualitative study. Nursing & Health Sciences 2013, 15 (1), 73-78.
